# Statistical bounds for entropic optimal transport: sample complexity and the central limit theorem

**Gonzalo Mena**
Harvard

**Jonathan Niles-Weed**
NYU

## Abstract

We prove several fundamental statistical bounds for entropic OT with the squared Euclidean cost between subgaussian probability measures in arbitrary dimension. First, through a new sample complexity result we establish the rate of convergence of entropic OT for empirical measures. Our analysis improves exponentially on the bound of Genevay et al. (2019) and extends their work to unbounded measures. Second, we establish a central limit theorem for entropic OT, based on techniques developed by Del Barrio and Loubes (2019). Previously, such a result was only known for finite metric spaces. As an application of our results, we develop and analyze a new technique for estimating the entropy of a random variable corrupted by gaussian noise.

## 1 Introduction

Optimal transport is an increasingly popular tool for the analysis of large data sets in high dimension, with applications in domain adaptation (Courty et al., 2014, 2017), image recognition (Li et al., 2013; Rubner et al., 2000; Sandler and Lindenbaum, 2011), and word embedding (Alvarez-Melis and Jaakkola, 2018; Grave et al., 2018). Its flexibility and simplicity have made it an attractive choice for practitioners and theorists alike, and its ubiquity as a machine learning tool continues to grow (see, e.g., Peyré et al., 2019; Kolouri et al., 2017, for surveys).

Much of the recent interest in optimal transport has been driven by algorithmic advances, chief among them the popularization of entropic regularization as a tool of solving large-scale OT problems quickly (Cuturi, 2013). Not only has this proposal been shown to yield near-linear-time algorithms for the original optimal transport problem (Altschuler et al., 2017), but it also appears to possess useful *statistical* properties which make it an attractive choice for machine learning applications (Rigollet and Weed, 2018; Genevay et al., 2017; Schiebinger et al., 2019; Montavon et al., 2016). For instance, in a recent breakthrough work, Genevay et al. (2019) established that even though the empirical version of standard OT suffers from the "curse of dimensionality" (see, e.g. Dudley, 1969), the empirical version of entropic OT always converges at the parametric $1/\sqrt{n}$ rate for compactly supported probability measures. This result suggests that entropic OT may be significantly more useful than unregularized OT for inference tasks when the dimension is large. However, obtaining rigorous guarantees for the performance of entropic OT in practice requires a more thorough understanding of its statistical behavior.

### 1.1 Summary of contributions

We prove new results on the relation between the population and empirical version of the entropic cost, that is, between $S(P,Q)$ and $S(P_n, Q_n)$ (defined in Section 1.2, below). These results give the first characterization of the large-sample behavior of entropic OT for unbounded probability measures in arbitrary dimension. Specifically, we obtain: **(i)** New sample complexity bounds on $E|S(P,Q) - S(P_n, Q_n)|$: first, we improve on the results of Genevay et al. (2019) by an exponential

factor, and then, extend these to unbounded measures (Section 2). **(ii)** A central limit theorem characterizing the fluctuations $S(P_n, Q_n) - ES(P_n, Q_n)$ when $P$ and $Q$ are subgaussian (Section 3). Such a central limit theorem was previously only known for probability measures supported on a finite number of points (Bigot et al., 2017; Klatt et al., 2018). We use completely different techniques, inspired by recent work of Del Barrio and Loubes (2019), to prove our theorem for general subgaussian distributions.

As an application of our results, we show how entropic OT can be used to shed new light on the entropy estimation problem for random variables corrupted by subgaussian noise (Section 4). This problem has gained recent interest in machine learning (Goldfeld et al., 2018, 2019) as a tool for obtaining a theoretically sound understanding of the *Information Bottleneck Principle* in deep learning (Tishby and Zaslavsky, 2015). We design and analyze a new estimator for this problem based on entropic OT.

Finally, we provide simulations which give empirical validation for our theoretical claims (Section 5).

## 1.2 Background and preliminaries

Let $P, Q \in \mathcal{P}(\mathbb{R}^d)$ be two probability measures and let $P_n$ and $Q_n$ be the empirical measures from the independent samples $\{X_i\}_{i \leq n} \sim P^n$ and $\{Y_i\}_{i \leq n} \sim Q^n$. We define the squared Wasserstein distance between $P$ and $Q$ (Villani, 2008) as follows:

$$W_2^2(P, Q) := \inf_{\pi \in \Pi(P,Q)} \left[ \int_{\mathcal{X} \times \mathcal{Y}} \frac{1}{2} \|x - y\|^2 \, \mathrm{d}\pi(x, y) \right], \tag{1}$$

where $\Pi(P, Q)$ is the set of all joint distributions with marginals equal to $P$ and $Q$, respectively. We focus on a entropy regularized version of the above cost (Cuturi, 2013; Peyré et al., 2019), defined as

$$S_\epsilon(P, Q) := \inf_{\pi \in \Pi(P,Q)} \left[ \int_{\mathcal{X} \times \mathcal{Y}} \frac{1}{2} \|x - y\|^2 \, \mathrm{d}\pi(x, y) + \epsilon H(\pi | P \otimes Q) \right], \tag{2}$$

where $H(\alpha | \beta)$ denotes the relative entropy between probability measures $\alpha$ and $\beta$ defiend by $\int \log \frac{d\alpha}{d\beta}(x) d\alpha(x)$ if $\alpha \ll \beta$ and $+\infty$ otherwise. By rescaling the measures $P$ and $Q$ and the regularization parameter $\epsilon$, it suffices to analyze the case $\epsilon = 1$, which we denote by $S(P, Q)$. Note that we have considered the squared cost $\frac{1}{2}\| \cdot \|^2$ in the definition of $S_\epsilon(P, Q)$, since most of our bounds heavily depend on this cost. However, more general costs $c(x, y)$ may be considered, and indeed some of our results (e.g. Proposition 4) are stated for more general $c(x, y)$. We leave a full analysis of the general case to future work.

The general theory of entropic OT (Csiszár, 1975) implies that $S(P, Q)$ possesses a dual formulation:

$$S(P, Q) = \sup_{f \in L_1(P), g \in L_1(Q)} \int f(x) \, \mathrm{d}P(x) + \int g(y) \, \mathrm{d}Q(y) - \int e^{f(x) + g(y) - \frac{1}{2}\|x - y\|^2} \, \mathrm{d}P(x)\mathrm{d}Q(y) + 1, \tag{3}$$

and that as long as $P$ and $Q$ have finite second moments, the supremum is attained at a pair of optimal potentials $(f, g)$ satisfying

$$\int e^{f(x) + g(y) - \frac{1}{2}\|x - y\|^2} \, \mathrm{d}Q(y) = 1 \quad P\text{-a.s.}, \qquad \int e^{f(x) + g(y) - \frac{1}{2}\|x - y\|^2} \, \mathrm{d}P(x) = 1 \quad Q\text{-a.s.} \tag{4}$$

Conversely, any $f \in L_1(P), g \in L_1(Q)$ satisfying (4) are optimal potentials.

We focus throughout on subgaussian probability measures. We say that a distribution $P \in \mathcal{P}(\mathbb{R}^d)$ is $\sigma^2$-subgaussian for $\sigma \geq 0$ if $E_P e^{\frac{\|X\|^2}{2d\sigma^2}} \leq 2$. By Jensen's inequality, if $E_P e^{\frac{\|X\|^2}{2d\sigma^2}} \leq C$ for any constant $C \geq 2$, then $P$ is $C\sigma^2$-subgaussian. Note that if $P$ is subgaussian, then $E_P e^{v^\top X} < \infty$ for all $v \in \mathbb{R}^d$. Conversely, standard results (see, e.g., Vershynin, 2018) imply that our definition is satisfied if $E_P e^{u^\top X} \leq e^{\|u\|^2 \sigma^2 / 2}$ for all $u \in \mathbb{R}^d$.

## 2 Sample complexity for the entropic transportation cost for general subgaussian measures

One rigorous statistical benefit of entropic OT is its *sample complexity*, i.e., the minimum number of samples required for the empirical entropic OT cost $S(P_n, Q_n)$ to be an accurate estimate of $S(P, Q)$. As noted above, unregularized OT suffers from the curse of dimensionality: in general, the Wasserstein distance $W_2^2(P_n, Q_n)$ converges to $W_2^2(P, Q)$ no faster than $n^{-1/d}$ for measures in $\mathbb{R}^d$. Strikingly, Genevay et al. (2019) established that the statistical performance of the entropic OT cost is significantly better. They show:[1]

**Theorem 1** (Genevay et al., 2019, Theorem 3). *Let $P$ and $Q$ be two probability measures on a bounded domain in $\mathbb{R}^d$ of diameter $D$. Then*

$$\sup_{P,Q} E_{P,Q}|S_\epsilon(P,Q) - S_\epsilon(P_n, Q_n)| \leq K_{D,d} \left(1 + \frac{1}{\epsilon^{\lfloor d/2 \rfloor}}\right) \frac{e^{D^2/\epsilon}}{\sqrt{n}}, \tag{5}$$

*where $K_{D,d}$ is a constant depending on $D$ and $d$.*

This impressive result offers powerful evidence that entropic OT converges significantly faster than its unregularized counterpart. The drawbacks of this result are that it applies only to bounded measures, and, perhaps more critically in applications, the rate scales *exponentially* in $D$ and $1/\epsilon$, even in dimension 1. Therefore, while the qualitative message of Theorem 1 is clear, it does not offer useful quantitative bounds as soon as the measure is unbounded or lies in a set of large diameter.

Our first theorem is a significant sharpening of Theorem 1. We first state it for the case where $\epsilon = 1$.

**Theorem 2.** *If $P$ and $Q$ are $\sigma^2$-subgaussian, then*

$$E_{P,Q}|S(P,Q) - S(P_n, Q_n)| \leq K_d(1 + \sigma^{\lceil 5d/2 \rceil + 6}) \frac{1}{\sqrt{n}}. \tag{6}$$

If we denote by $P^\epsilon$ and $Q^\epsilon$ the pushforwards of $P$ and $Q$ under the map $x \mapsto \epsilon^{-1/2}x$, then it is easy to see that

$$S_\epsilon(P,Q) = \epsilon S(P^\epsilon, Q^\epsilon).$$

We immediately obtain the following corollary.

**Corollary 1.** *If $P$ and $Q$ are $\sigma^2$-subgaussian, then*

$$E_{P,Q}|S_\epsilon(P,Q) - S_\epsilon(P_n, Q_n)| \leq K_d \cdot \epsilon \left(1 + \frac{\sigma^{\lceil 5d/2 \rceil + 6}}{\epsilon^{\lceil 5d/4 \rceil + 3}}\right) \frac{1}{\sqrt{n}}.$$

If we compare Corollary 1 with Theorem 1, we note that the polynomial prefactor in Corollary 1 has higher degree than the one in Theorem 1, pointing to a potential weakness of our bound. On the other hand, the exponential dependence on $D^2/\epsilon$ has completely disappeared. Moreover, the brittle quantity $D$, finite only for compactly supported measures, has been replaced by the more flexible subgaussian variance proxy $\sigma^2$.

The improvements in Theorem 2 are obtained via two different methods. First, a simple argument allows us to remove the exponential term and bound the desired quantity by an empirical process, as in Genevay et al. (2019). Much more challenging is the extension to measures with unbounded support. The proof technique of Genevay et al. (2019) relies on establishing uniform bounds on the derivatives of the optimal potentials, but this strategy cannot succeed if the support of $P$ and $Q$ is not compact. We therefore employ a more careful argument based on controlling the Hölder norms of the optimal potentials on compact sets. A chaining bound completes our proof.

In Proposition 1 below (whose proof we defer to the supplement) we show that if $(f, g)$ is a pair of optimal potentials for $\sigma^2$-subgaussian distributions $P$ and $Q$, then we may control the size of $f$ and its derivatives.

**Proposition 1.** *Let $P$ and $Q$ be $\sigma^2$-subgaussian distributions. There exist optimal dual potentials $(f, g)$ for $P$ and $Q$ such that for any multi-index $\alpha$ with $|\alpha| = k$,*

$$|D^\alpha(f - \frac{1}{2}\|\cdot\|^2)(x)| \leq C_{k,d} \begin{cases} 1 + \sigma^4 & k = 0 \\ \sigma^k(\sigma + \sigma^2)^k & otherwise, \end{cases} \tag{7}$$

*if $\|x\| \leq \sqrt{d}\sigma$, and*

$$|D^\alpha(f - \frac{1}{2}\|\cdot\|^2)(x)| \leq C_{k,d} \begin{cases} 1 + (1+\sigma^2)\|x\|^2 & k = 0 \\ \sigma^k(\sqrt{\sigma\|x\|} + \sigma\|x\|)^k & otherwise, \end{cases} \tag{8}$$

*if $\|x\| > \sqrt{d}\sigma$, where $C_{k,d}$ is a constant depending only on $k$ and $d$.*

We denote by $\mathcal{F}_\sigma$ the set of functions satisfying the bounds (7) and (8). The following proposition shows that it suffices to control an empirical process indexed by this set.

**Proposition 2.** *Let $P$, $Q$, and $P_n$ be $\tilde{\sigma}^2$-subgaussian distributions, for a possibly random $\tilde{\sigma} \in [0, \infty)$. Then*

$$|S(P_n, Q) - S(P, Q)| \leq 2 \sup_{u \in \mathcal{F}_{\tilde{\sigma}}} |E_P u - E_{P_n} u|. \tag{9}$$

*Proof.* We define the operator $\mathcal{A}^{\alpha,\beta}(u, v)$ for the pair of probability measures $(\alpha, \beta)$ and functions $(u, v) \in L_1(\alpha) \otimes L_1(\beta)$ as:

$$\mathcal{A}^{\alpha,\beta}(u, v) = \int u(x)\, \mathrm{d}\alpha(x) + \int v(y)\, \mathrm{d}\beta(y) - \int e^{u(x)+v(y)-\frac{1}{2}\|x-y\|^2}\, \mathrm{d}\alpha(x)d\beta(y) + 1.$$

Denote by $(f_n, g_n)$ a pair of optimal potentials for $(P_n, Q)$ and $(f, g)$ for $(P, Q)$, respectively. By Proposition A.1 in the supplement, we can choose smooth optimal potentials $(f, g)$ and $(f_n, g_n)$ so that the condition (4) holds for all $x, y \in \mathbb{R}^d$. Proposition 1 shows that $f, f_n \in \mathcal{F}_{\tilde{\sigma}}$.

Strong duality implies that $S(P, Q) = \mathcal{A}^{P,Q}(f, g)$ and $S(P_n, Q) = \mathcal{A}^{P_n,Q}(f_n, g_n)$. Moreover, by the optimality of $(f, g)$ and $(f_n, g_n)$ for their respective dual problems, we obtain

$$\mathcal{A}^{P,Q}(f_n, g_n) - \mathcal{A}^{P_n,Q}(f_n, g_n) \leq \mathcal{A}^{P,Q}(f, g) - \mathcal{A}^{P_n,Q}(f_n, g_n) \leq \mathcal{A}^{P,Q}(f, g) - \mathcal{A}^{P_n,Q}(f, g).$$

We conclude that

$$\begin{aligned} |S(P, Q) - S(P_n, Q)| &= |\mathcal{A}^{P,Q}(f, g) - \mathcal{A}^{P_n,Q}(f_n, g_n)| \\ &\leq |\mathcal{A}^{P,Q}(f, g) - \mathcal{A}^{P_n,Q}(f, g)| + |\mathcal{A}^{P,Q}(f_n, g_n) - \mathcal{A}^{P_n,Q}(f_n, g_n)|. \end{aligned}$$

It therefore suffices to bound the differences $|\mathcal{A}^{P,Q}(f, g) - \mathcal{A}^{P_n,Q}(f, g)|$ and $|\mathcal{A}^{P,Q}(f_n, g_n) - \mathcal{A}^{P_n,Q}(f_n, g_n)|$.

Upon defining $h(x) := \int e^{g(y)-\frac{1}{2}\|x-y\|^2}\,\mathrm{d}Q(y)$ we have

$$\mathcal{A}^{P,Q}(f, g) - \mathcal{A}^{P_n,Q}(f, g) = \left(\int f(x)(\mathrm{d}P(x) - \mathrm{d}P_n(x))\right) + \left(\int e^{f(x)}h(x)(\mathrm{d}P(x) - \mathrm{d}P_n(x))\right).$$

Since $(f, g)$ satisfy $e^{f(x)}h(x) = 1$ for all $x \in \mathbb{R}^d$, the second term above vanishes. Therefore

$$|\mathcal{A}^{P,Q}(f, g) - \mathcal{A}^{P_n,Q}(f, g)| = \left|\int f(x)(\mathrm{d}P(x) - \mathrm{d}P_n(x))\right| \leq \sup_{u \in \mathcal{F}_{\tilde{\sigma}}} \left|\int u(x)(\mathrm{d}P(x) - \mathrm{d}P_n(x))\right|.$$

Analogously,

$$|\mathcal{A}^{P,Q}(f_n, g_n) - \mathcal{A}^{P_n,Q}(f_n, g_n)| \leq \sup_{u \in \mathcal{F}_{\tilde{\sigma}}} \left|\int u(x)(\mathrm{d}P(x) - \mathrm{d}P_n(x))\right|.$$

This proves the claim. $\square$

Proposition 2 can be extended to apply to simultaneously varying $P_n$ and $Q_n$.

**Corollary 2.** *Let $P$, $Q$, $P_n$, and $Q_n$ be $\tilde{\sigma}^2$-subgaussian distributions, where $\tilde{\sigma} \in [0, \infty)$ is possibly random. Then*

$$|S(P_n, Q_n) - S(P, Q)| \lesssim \sup_{u \in \mathcal{F}_{\tilde{\sigma}}} \left|\int u(x)(\mathrm{d}P(x) - \mathrm{d}P_n(x))\right| + \sup_{u \in \mathcal{F}_{\tilde{\sigma}}} \left|\int u(x)(\mathrm{d}Q(x) - \mathrm{d}Q_n(x))\right|$$

*almost surely.*

*Proof.* By the triangle inequality,

$$|S(P_n, Q_n) - S(P, Q)| \le |S(P, Q) - S(P_n, Q)| + |S(P_n, Q) - S(P_n, Q_n)|. \qquad (10)$$

Since $P$, $Q$, $P_n$, and $Q_n$ are all $\tilde{\sigma}^2$-subgaussian, Proposition 2 can be applied to both terms. $\qquad \square$

The majority of our work goes into bounding the resulting empirical process. Let $s \ge 2$. Fix a constant $C_{s,d}$ and denote by $\mathcal{F}^s$ the set of functions satisfying

$$|f(x)| \le C_{s,d}(1 + \|x\|^2) \qquad (11)$$
$$|D^\alpha f(x)| \le C_{s,d}(1 + \|x\|^s) \qquad \forall \alpha : |\alpha| \le s. \qquad (12)$$

Proposition 1 establishes that if $C_{s,d}$ is large enough, then $\frac{1}{1+\sigma^{3s}} f \in \mathcal{F}^s$ for all $f \in \mathcal{F}_\sigma$.

The key result is the following covering number bound, whose proof we defer to the supplement. Denote by $N(\varepsilon, \mathcal{F}^s, L_2(P_n))$ the covering number with respect to the (random) metric $L_2(P_n)$ defined by $\|f\|_{L_2(P_n)} = \left(\frac{1}{n} \sum_{i=1}^n f(X_i)^2\right)^{1/2}$.

**Proposition 3.** *Let $s = \lceil d/2 \rceil + 1$. If $P$ is $\sigma^2$-subgaussian and $P_n$ is an empirical distribution, then there exists a random variable $L$ depending on the sample $X_1, \ldots, X_n$ satisfying $EL \le 2$ such that*

$$\log N(\varepsilon, \mathcal{F}^s, L_2(P_n)) \le C_d L^{d/2s} \varepsilon^{-d/s}(1 + \sigma^{2d}),$$

*and*

$$\max_{f \in \mathcal{F}^s} \|f\|_{L_2(P_n)}^2 \le C_d(1 + L\sigma^4).$$

We can now prove Theorem 2.

*Proof of Theorem 2.* Let $\tilde{\sigma}$ be the infimum over all $\tau > 0$ such that $P$, $Q$, $P_n$, and $Q_n$ are all $\tau^2$-subgaussian. By Lemma A.2 in the supplement, $\tilde{\sigma}$ is finite almost surely.

By Corollary 2,

$$E_{P,Q}|S(P, Q) - S(P_n, Q_n)| \lesssim E \sup_{u \in \mathcal{F}_{\tilde{\sigma}}} \left| \int u(x)(\mathrm{d}P(x) - \mathrm{d}P_n(x)) \right|$$
$$+ E \sup_{u \in \mathcal{F}_{\tilde{\sigma}}} \left| \int u(x)(\mathrm{d}Q(x) - \mathrm{d}Q_n(x)) \right|.$$

We will show how to bound the first term, and the second will follow in exactly the same way.

For any set of functions $\mathcal{F}$, we write $\|P - P_n\|_{\mathcal{F}} = \sup_{u \in \mathcal{F}} (\int u(x)(\mathrm{d}P(x) - \mathrm{d}P_n(x)))$. Recall that, for $s = \lceil d/2 \rceil + 1$, if $u \in \mathcal{F}_{\tilde{\sigma}}$ then $\frac{1}{1+\tilde{\sigma}^{3s}} u \in \mathcal{F}^s$. Therefore

$$E\|P - P_n\|_{\mathcal{F}_\sigma} \le E(1 + \tilde{\sigma}^{3s})\|P - P_n\|_{\mathcal{F}^s}$$
$$\le (E(1 + \tilde{\sigma}^{3s})^2)^{1/2}(E\|P - P_n\|_{\mathcal{F}^s}^2)^{1/2}.$$

Then by Giné and Nickl (2016, Theorem 3.5.1 and Exercise 2.3.1), we have

$$E\|P - P_n\|_{\mathcal{F}^s}^2 \lesssim \frac{1}{n} E \left( \int_0^{\sqrt{\max_{f \in \mathcal{F}^s} \|f\|_{L_2(P_n)}^2}} \sqrt{\log 2N(\tau, \mathcal{F}^s, L_2(P_n))} \, \mathrm{d}\tau \right)^2$$

$$\le C_d \frac{1}{n} E \left( \int_0^{C_d\sqrt{(1+L\sigma^4)}} \sqrt{1 + L^{d/2s}\tau^{-d/s}(1 + \sigma^{2d})} \, \mathrm{d}\tau \right)^2$$

$$\le C_d \frac{1}{n}(1 + \sigma^{2d}) E \left( \int_0^{C_d\sqrt{(1+L\sigma^4)}} L^{d/4s}\tau^{-d/2s} \, \mathrm{d}\tau \right)^2$$

$$\le C_d \frac{1}{n}(1 + \sigma^{2d}) E \left[ (1 + L\sigma^4)^{1-d/2s} \right],$$

where in the last step we have used that $d/2s < 1$ so that $\tau^{-d/2s}$ is integrable in a neighborhood of the origin. Applying the bound on $EL$ yields that this expression is bounded by $C_d(1 + \sigma^{2d+4})\frac{1}{n}$.

Lemma A.4 in the supplement shows that $E\tilde{\sigma}^{2k} \leq C_k\sigma^{2k}$ for all positive integers $k$. Combining these bounds yields

$$E\|P - P_n\|_{\mathcal{F}_\sigma} \leq C_d(1 + \sigma^{3s})(1 + \sigma^{d+2})\frac{1}{\sqrt{n}},$$

as desired. □

## 3 A central limit theorem for entropic OT

The results of Section 2 show that, for general subgaussian measures, the empirical quantity $S(P_n, Q_n)$ converges to $S(P, Q)$ in expectation at the parametric rate. However, in order to use entropic OT for rigorous statistical inference tasks, much finer control over the deviations of $S(P_n, Q_n)$ is needed, for instance in the form of asymptotic distributional limits. In this section, we accomplish this goal by showing a central limit theorem (CLT) for $S(P_n, Q_n)$, valid for any subgaussian measures.

Bigot et al. (2017) and Klatt et al. (2018) have shown CLTs for entropic OT when the measures lie in a *finite* metric space (or, equivalently, when $P$ and $Q$ are finitely supported). Apart from being restrictive in practice, these results do not shed much light on the general situation because OT on finite metric spaces behaves quite differently from OT on $\mathbb{R}^d$.[2] Very recently, distributional limits for general measures possessing $4 + \delta$ moments have been obtained for unregularized OT by Del Barrio and Loubes (2019). Our proof follows their approach.

We prove the following.

**Theorem 3.** *Let $X_1, \ldots X_n \sim P$ be an i.i.d. sequence, and denote by (f,g) the optimal potentials in* (4)*. If $P$ is subgaussian, then*

$$\sqrt{n}\left(S(P_n, Q) - E(S(P_n, Q))\right) \xrightarrow{\mathcal{D}} \mathcal{N}\left(0, \mathrm{Var}_P(f(X))\right), \tag{13}$$

*and*

$$\lim_{n \to \infty} n \, \mathrm{Var}(S(P_n, Q)) = \mathrm{Var}_P(f(X)). \tag{14}$$

*Likewise, let $X_1, \ldots, X_n \sim P$ and $Y_1, \sim Y_m \sim Q$ are two i.i.d. sequences independent of each other. Assume $P$ and $Q$ are both subgaussian. Denote $\lambda := \lim_{m,n \to \infty} \frac{n}{m+n} \in (0, 1)$.*

*Then*

$$\sqrt{\frac{mn}{m+n}}\left(S(P_n, Q_m) - E(S(P_n, Q_m))\right) \xrightarrow{\mathcal{D}} \mathcal{N}\left(0, (1 - \lambda) \mathrm{Var}_P(f(X_1)) + \lambda \mathrm{Var}_Q(g(Y_1))\right), \tag{15}$$

*and*

$$\lim_{m,n \to \infty} \frac{mn}{m+n} \mathrm{Var}(S(P_n, Q_m)) = (1 - \lambda) \mathrm{Var}_P(f(X)) + \lambda \mathrm{Var}_Q(g(Y)). \tag{16}$$

The proof is deeply inspired by the method developed in Del Barrio and Loubes (2019) for the squared Wasserstein distance, and we roughly follow the same strategy.

*Proof of Theorem 3.* The proof, in the one-sample case, proceeds as follows:

(a) In Proposition A.2 we show that the optimal potentials for $(P_n, Q)$ convergence to optimal potentials for $(P, Q)$ uniformly on compact sets.

(b) Letting $R_n := S(P_n, Q) - \int f(x)\mathrm{d}P_n(x)$, we show in Proposition A.3 in the supplement, that this uniform convergence implies that $\lim_{n \to \infty} n \mathrm{Var}(R_n) = 0$.

(c) The above convergence indicates $S(P_n, Q)$ can be approximated by the linear quantity $\int f(x)\mathrm{d}P_n$. Then, (13) and (14) are simply the limit statements (in distribution and $L^2$, respectively) applied to this linearization.

We omit the proof of the two-sample case as the changes to the argument (see Theorem 3.3. in Del Barrio and Loubes (2019), for the squared Wasserstein distance) adapt in a straightforward way to the entropic case. □

# 4 Application to entropy estimation

In this section, we give an application of entropic OT to the problem of entropy estimation. First, in Proposition 4 we establish a new relation between entropic OT and the differential entropy of the convolution of two measures. Then, as a corollary of this and the previous sections results we prove Theorem 4, stating that entropic OT provides us with a novel estimator for the differential entropy of the (independent) sum of a subgaussian random variable and a gaussian random variable, and for which performance guarantees are available.

Throughout this section $\nu$ denotes a translation invariant measure. Whenever $P$ has a density $p$ with respect to $\nu$, we define its $\nu$-differential entropy as $h(P) := -\int p(x) \log p(x) d\nu(x) = -H(P|\nu)$.

The following proposition links the differential entropy of a convolution with the entropic cost.

**Proposition 4.** *Let $\Phi_g$ be the measure with $\nu$-density $\phi_g(y) = Z_g^{-1} e^{-g(y)}$ for a smooth $g$ ($Z_g$ is the normalizing constant), and define $Q = P * \Phi_g$, with $P \in \mathcal{P}(\mathbb{R}^d)$ arbitrary. The $\nu$-density of $Q$, $q(y)$, satisfies*

$$q(y) = \int \phi_g(y - x) dP(x) = \int Z_g^{-1} e^{-g(y-x)} dP(x).$$

*Consider the cost function $c(x,y) := g(x-y)$ (not necessarily quadratic). Then, the optimal entropic transport cost and differential entropy are linked through*

$$h(P * \Phi_g) = S(P, P * \Phi_g) + \log(Z_g). \tag{17}$$

*Proof.* Define a more general entropic transportation cost involving the generic $c$ and probability measures $\alpha, \beta$ [3]:

$$S^{\alpha \otimes \beta}(P, Q) := \inf_{\pi \in \Pi(P,Q)} \left[ \int c(x,y) d\pi(x,y) + H(\pi|\alpha \otimes \beta) \right]. \tag{18}$$

Observe we may re-write (18) as

$$
\begin{aligned}
S^{\alpha \otimes \beta}(P, Q) &= \inf_{\pi \in \Pi(P,Q)} \left[ \int_{\mathcal{X} \times \mathcal{Y}} c(x,y) d\pi(x,y) + H(\pi|P \otimes Q) \right] + H(P \otimes Q | \alpha \otimes \beta) \\
&= S(P, Q) + H(P \otimes Q | \alpha \otimes \beta).
\end{aligned} \tag{19}
$$

Additionally, it can be verified an alternative representation for (18) is the following

$$S^{\alpha \otimes \beta}(P, Q) = \inf_{\pi \in \Pi(P,Q)} H\left( \pi \Big| Z^{-1} e^{-c} \alpha \otimes \beta \right) - \log(Z), \tag{20}$$

where $Z$ is the number making $\Lambda := Z^{-1} e^{-c} \alpha \otimes \beta$ a *bona fide* probability measure.

Now, take $\alpha = P$, $\beta = \nu$ and $Q = P * \Phi_g$ in the above expressions. For these choices we have $Z = Z_g$. Indeed, by the translation invariance of $\nu$, we have

$$
\begin{aligned}
Z = \iint e^{-c(x,y)} dP(x) d\nu(y) &= \int \left( \int e^{-g(y-x)} d\nu(y) \right) dP(x) \\
&= \int \left( \int e^{-g(y)} d\nu(y) \right) dP(x) \\
&= \int Z_g dP(x) = Z_g.
\end{aligned}
$$

Then, $d\Lambda(x,y) = \mathrm{d}P(x)\phi_g(y-x)d\nu(y)$, and by marginalization we deduce $\Lambda \in \Pi(P, P * \Phi_g)$. Therefore, the right side of (20) equals $H(\Lambda|\Lambda) - \log Z_g = -\log Z_g$. Finally, we combine (19) and (20) to obtain

$$-\log Z_g = S(P, P * \Phi_g) + H\left(P \otimes (P * \Phi_g) \,|\, P \otimes \nu\right),$$

and achieve the final conclusion after noting that

$$H\left(P \otimes (P * \Phi_g) \,|\, P \otimes \nu\right) = H(P|P) + H\left(P * \Phi_g | \nu\right) = H\left(P * \Phi_g | \nu\right) = -h(P * \Phi_g).$$

$\square$

Now we can state the following theorem.

**Theorem 4.** *Let $P$ be subgaussian, and $\Phi_g = \mathcal{N}(0, \epsilon I_d)$. Denote $Q = P * \Phi_g$ the distribution of the sum of an independent samples from $P$ and $\Phi_g$, and define the plug in estimator $\hat{h}(Q) = S(P_n, Q_m) + \log Z_g$ where $P_n$ and $Q_m$ are independent samples from $P$ and $Q$. Then,*

*(a) If $m = n$,*

$$\sup_P E_P |\hat{h}(Q) - h(Q)| \le O\left(\frac{1}{\sqrt{n}}\right).$$

*(b) The limit*

$$\sqrt{\frac{mn}{m+n}} \left(\hat{h}(Q) - E(\hat{h}(Q))\right) \xrightarrow{\mathcal{D}} \mathcal{N}\left(0, \lambda \operatorname{Var}_Q(\log q(Y))\right) \qquad (21)$$

*holds, where $\lambda = \lim_{m,n\to\infty} \frac{n}{m+n}$. Moreover, $\lim_{m,n\to\infty} \frac{mn}{m+n} \operatorname{Var}(\hat{h}(Q)) = \lambda \operatorname{Var}_Q(\log q(Y))$.*

*Proof.* (a) is a simple re-statement of Theorem 2 in the light of Proposition 4. (b) is a re-statement of Theorem 3, after noting in this case the optimal potentials are $(f, g) = (-\log Z_g, -\log q)$. $\square$

The rate $1/\sqrt{n}$ in Theorem 4 is also achieved by a different estimator proposed by Goldfeld et al. (2019) (see also Weed, 2018), but this estimator lacks distributional limits.

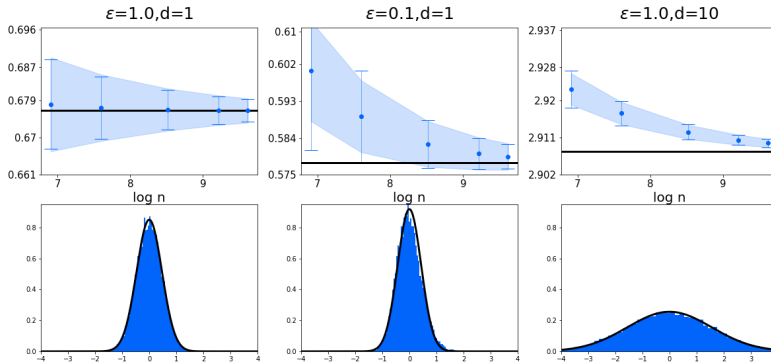

Figure 1: Top row: $ES(P_n, Q_n)$ as a function of $n \in \{1e3, 2e3, 5e3, 1e4, 1.5e4\}$, computed from $16{,}000$ repetitions for each value of $n$. The shading corresponds to one standard deviation of $S(P_n, Q_n) - ES(P_n, Q_n)$, assuming the asymptotics of Theorem 3 are valid. Error bars are one sample standard deviation long on each side. Both $x$ and $y$ axes are in logarithmic scale. Bottom row: histograms of $\sqrt{\frac{nn}{n+n}} \left(S(P_n, Q_n) - ES(P_n, Q_n)\right)$ when $n = 1.5e4$. Ground truth (numerical integration) is shown with black solid lines.

# 5  Empirical results

We provide empirical evidence supporting and illustrating our theoretical findings. We focus on the entropy estimation problem because there are closed form expressions for the potentials (see Theorem 4), and because it allows a comparison with the estimator studied in (Goldfeld et al., 2019) [4].

Specifically, consider $X \sim P = \frac{1}{2}\left(\mathcal{N}(1_d, I_d) + \mathcal{N}(-1_d, I_d)\right)$, the mixture of the gaussians centered at $1_d := (1, \ldots, 1)$ and $-1_d$. We aim to estimate the entropy of the new mixture $Q = P * \Phi_g$.

Figure 1, top, shows the convergence of $ES(P_n, Q_n)$ to $S(P, Q)$. Consistent with the bound in Theorem 2 and Corollary 1, $S(P_n, Q_n)$ is a worse estimator for $S(P, Q)$ when $d$ is large or the regularization parameter is small. We also plot the predicted (shading) and actual (bars) fluctuations of $S(P_n, Q_n)$ around its mean. Though the CLT holds only in the asymptotic limit, these experiments reveal that the empirical fluctuations in the finite-$n$ regime are broadly consistent with the predictions of the CLT. Figure 1, bottom, shows that the empirical distribution of the rescaled fluctuations is an excellent match for the predicted normal distribution.

In Figure 2 we compare the performance between entropic OT-based estimators from Theorem 4 and $\hat{h}_{\mathrm{m.g.}}(Q)$, the one from (Goldfeld et al., 2019), where $h(P * \Phi_g)$ is estimated as the entropy of the mixture of gaussians $P_n * \Phi_g$, in turn approximated by Monte Carlo integration. We consider two OT-based estimators, $\hat{h}_{\mathrm{ind}}(Q)$ where $P_n, Q_n$ are completely independent (i.e., the one used for Figure 1), and $\hat{h}_{\mathrm{paired}}(Q)$ where samples $Q_n$ are drawn by adding gaussian noise to $P_n$. Observe that our sample complexity and CLT results are only available for $\hat{h}_{\mathrm{ind}}(Q)$.

Results show a clear pattern of dominance, with $E\hat{h}_{\mathrm{paired}}(Q)$ achieving the fastest convergence. The main caveat is the extra memory cost: while $\hat{h}_{\mathrm{m.g.}}(Q)$ can be computed sequentially with each operation requiring $O(n)$ memory, in the most naive implementation (used here) both $\hat{h}_{\mathrm{paired}}(Q), \hat{h}_{\mathrm{ind}}(Q)$ demand $O(n^2)$ space for storing the matrix $D_{i,j} = e^{-||x_i - y_j||^2/2\epsilon}$, to which the Sinkhorn algorithm is applied. This memory requirement might be alleviated with the use of stochastic methods (Genevay et al., 2016; Bercu and Bigot, 2018).

We leave for future work both the implementation of more scalable methods for entropic OT, and a detailed theoretical analysis of different entropic OT-based estimators (e.g. $\hat{h}_{\mathrm{paired}}(Q)$ v.s. $\hat{h}_{\mathrm{ind}}(Q)$) that may bring about a better understanding of their observed substantial differences. Additionally, in future work we will explore extensions of our results beyond the subgaussian case, and provide lower bounds as in Goldfeld et al. (2019).

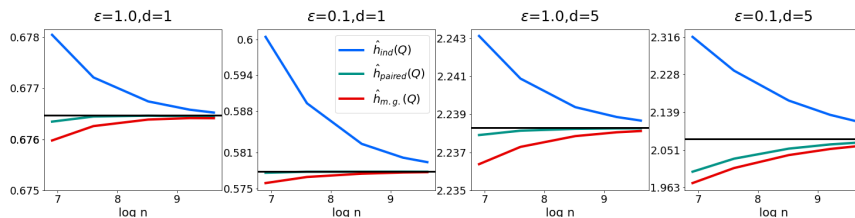

Figure 2: Comparison between $E\hat{h}_{\mathrm{ind}}(Q), E\hat{h}_{\mathrm{paired}}(Q), E\hat{h}_{\mathrm{m.g.}}(Q)$. Details are the same as in Figure 1.

## Footnotes

[1] We have specialized their result to the squared Euclidean cost.

[2]A thorough discussion of the behavior of unregularized OT for finitely supported measures can be found in Sommerfeld and Munk (2018) and Weed and Bach (2018).

[3]Notice $\alpha \otimes \beta$ need not be probability measures for the relative entropy $H(\cdot | \alpha \otimes \beta)$ to make sense. In Léonard (2014) it is argued it suffices that this product is $\sigma$-finite.

[4]We don't present comparisons with the recent estimator presented in Berrett et al. (2019). This general purpose estimator is $\sqrt{n}$−consistent and a CLT is available without a centering constant (as our $ES(P_n, Q_n)$). However, empirical results show the one in Goldfeld et al. (2019) performs much better.

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
