[Supplementary Material · neurips_2019_supplement.pdf]

# Supplement to "Statistical bounds for entropic optimal transport: sample complexity and the central limit theorem"

**Gonzalo Mena**
Harvard

**Jonathan Niles-Weed**
NYU

Throughout the supplement, the symbol $C$ will be used to indicate an unspecified positive constant whose value may change from line to line. Subscripts will be used to indicate if $C$ depends on any other parameters.

## A    Supplementary results

**Lemma A.1.** *If $P$ is $\sigma^2$ subgaussian, then*

$$E_P\|X\|^{2k} \leq (2d\sigma^2)^k k!$$

*for all nonnegative integers $k$, and*

$$E_P e^{v \cdot X} \leq E_P e^{\|v\|\|X\|} \leq 2e^{\frac{d\sigma^2}{2}\|v\|^2}$$

*for all $v \in \mathbb{R}^d$.*

*Proof.* For the first claim, it suffices to take expectations of both sides of the inequality $\frac{\|X\|^{2k}}{(2d\sigma^2)^k k!} \leq e^{\frac{\|X\|^2}{2d\sigma^2}} - 1$ and use the assumption that $P$ is $\sigma^2$-subgaussian. To prove the second claim, we use the inequality $v \cdot X \leq \|v\|\|X\| \leq \frac{d\sigma^2}{2}\|v\|^2 + \frac{1}{2d\sigma^2}\|X\|^2$ and apply subgaussianity. $\square$

**Proposition A.1.** *Let $P$ and $Q$ be two $\sigma^2$-subgaussian distributions. Then there exist smooth optimal potentials $(f, g)$ for $S(P, Q)$ such that*

$$-d\sigma^2(1 + \frac{1}{2}(\|x\| + \sqrt{2d}\sigma)^2) - 1 \leq f(x) \leq \frac{1}{2}(\|x\| + \sqrt{2d}\sigma)^2$$

$$-d\sigma^2(1 + \frac{1}{2}(\|y\| + \sqrt{2d}\sigma)^2) - 1 \leq g(y) \leq \frac{1}{2}(\|y\| + \sqrt{2d}\sigma)^2$$

*and the dual optimality conditions* (4) *hold for all $x, y \in \mathbb{R}^d$.*

*Proof.* Let $(f_0, g_0)$ be any pair of optimal potentials. Since $(f_0 + K, g_0 - K)$ also satisfy the optimality conditions and $f_0 \in L_1(P)$ and $g_0 \in L_1(Q)$, we can assume without loss of generality that $E_P f_0(X) = E_Q g_0(Y) = \frac{1}{2}S(P, Q) \geq 0$. We define

$$f(x) = -\log \int e^{g_0(y) - \frac{1}{2}\|x-y\|^2} \, dQ(y)$$

$$g(y) = -\log \int e^{f(x) - \frac{1}{2}\|x-y\|^2} \, dP(x),$$

for all $x, y \in \mathbb{R}^d$.

We need to check that these integrals are well defined. First, Jensen's inequality implies

$$g_0(y) = -\log \int e^{f_0(x) - \frac{1}{2}\|x-y\|^2} \, \mathrm{d}P(x)$$

$$\leq -E_P f_0(X) + \frac{1}{2} E_P \|X - y\|^2$$

$$\leq \frac{1}{2} E_P \|X - y\|^2$$

for $Q$-a.e. y. Therefore

$$e^{g_0(y) - \frac{1}{2}\|x-y\|^2} \leq e^{\frac{1}{2}E_P\|X-y\|^2 - \frac{1}{2}\|x-y\|^2}$$

for $Q$-a.e. y. By Lemma A.1, $E_P\|X\|^2 \leq 2d\sigma^2$, which implies that $e^{g_0(y) - \frac{1}{2}\|x-y\|^2}$ is dominated by $e^{d\sigma^2 + (\|x\| + \sqrt{2}d\sigma)\|y\|}$. Subgaussianity implies

$$\int e^{d\sigma^2 + (\|x\| + \sqrt{2}d\sigma)\|y\|} \, \mathrm{d}Q(y) \leq 2 e^{d\sigma^2(1 + \frac{1}{2}(\|x\| + \sqrt{2}d\sigma)^2} < \infty$$

Therefore $f(x)$ is well defined for all $x \in \mathbb{R}^d$. The same argument used to bound $g_0$ holds for $f$ as well, which implies that $g$ is also well defined. Therefore our definitions of $f$ and $g$ are valid on the whole space, and moreover the claimed lower bounds on $f$ and $g$ hold. Jensen's inequality combined with the inequalities $E_Q g_0(Y) \geq 0$ and $E_P f(X) \geq 0$ yield the upper bounds. The smoothness of $f$ and $g$ follows from an easy application of dominated convergence.

We now show that $(f, g)$ are optimal potentials. By construction $\int e^{f(x) + g(y) - \frac{1}{2}\|x-y\|^2} \, \mathrm{d}P(x) = 1$ for all $y \in \mathbb{R}^d$. Now, note that

$$\int e^{f(x) + g(y) - \frac{1}{2}\|x-y\|^2} \, \mathrm{d}P(x)\mathrm{d}Q(y) = \int e^{f(x) + g_0(y) - \frac{1}{2}\|x-y\|^2} \, \mathrm{d}P(x)\mathrm{d}Q(y)$$

$$= \int e^{f_0(x) + g_0(y) - \frac{1}{2}\|x-y\|^2} \, \mathrm{d}P(x)\mathrm{d}Q(y) \, .$$

Jensen's inequality yields

$$\int (f - f_0)(x) \, \mathrm{d}P(x) + \int (g - g_0)(y) \, \mathrm{d}Q(y) \geq -\log \int e^{f_0(x) - f(x)} \, \mathrm{d}P(x) - \log \int e^{g_0(y) - g(y)} \, \mathrm{d}Q(y)$$

$$= -\log \int e^{f_0(x) + g_0(y) - \frac{1}{2}\|x-y\|^2} \, \mathrm{d}P(x)\mathrm{d}Q(y)$$

$$- \log \int e^{f(x) + g_0(y) - \frac{1}{2}\|x-y\|^2} \, \mathrm{d}P(x)\mathrm{d}Q(y)$$

$$= 0 \, .$$

Since $(f_0, g_0)$ maximizes (3), so does $(f, g)$. Therefore $(f, g)$ are optimal potentials. In particular, this implies that $\int (g - g_0)(y) \, \mathrm{d}Q(y) = \log \int e^{g_0(y) - g(y)} \, \mathrm{d}Q(y)$, and hence $g = g_0$ $Q$-almost surely by the strict concavity of the logarithm function. We obtain that $\int e^{f(x) + g(y) - \frac{1}{2}\|x-y\|^2} \, \mathrm{d}Q(y) = \int e^{f(x) + g_0(y) - \frac{1}{2}\|x-y\|^2} \, \mathrm{d}Q(y) = 1$ for all $x \in \mathbb{R}^d$. $\square$

**Proposition A.2.** *Let $P_n, Q_n$ be empirical measures, $P$ and $Q$ both assumed subgaussian. There exist $(f_n, g_n)$ optimal potentials for $(P_n, Q_n)$ such that $(f_n, g_n)$ converges uniformly in compacts to optimal potentials $(f, g)$ for $P$ and $Q$.*

*Proof.* The proof is inspired by Feydy et al. (2019) and we divide it in two steps:

Step 1 By using the following extended version of the Arzela-Ascoli theorem we find a convergent subsequence: suppose $h_n$ is a sequence of functions in $\mathbb{R}^d$ satisfying

(a) Local equicontinuity: for each $x_0 \in \mathbb{R}^d$ and $\epsilon > 0$, there is a $\delta > 0$ such that

$$\|x - x_0\| < \delta \quad \text{implies} \quad |h_n(x) - h_n(x_0)| < \epsilon \quad \text{for all } n$$

(b) Pointwise boundedness: for each $x$, the sequence $h_n(x)$ is bounded.

Then, there exist a subsequence $h_{n_j}$ that converges uniformly on compacts to a continuous function $h$.

Step 2 We prove the limit functions are optimal for $(P, Q)$ and conclude the entire sequence converges by a uniqueness argument.

*Proof of Step 1*: By Lemma A.2, there exists a (random) $\sigma^2$ such that the measures $\{P_n\}$ are uniformly $\sigma^2$-subgaussian. We choose $(f_n, g_n)$ and $(f, g)$ as in Proposition A.1.

By Proposition A.1, $(f_n, g_n)$ are pointwise bounded by a quantity independent of $n$. Likewise, Proposition 1 implies that the derivatives of $f_n$ and $g_n$ are also pointwise bounded, which implies local equicontinuity.

We conclude for a certain subsequence $n_j$, $(f_{n_j}, g_{n_j})$ converges to some $(f_\infty, g_\infty)$.

*Proof of Step 2:* It is easy to verify (by Jensen's inequality and dominated convergence) that Proposition 11 in Feydy et al. (2019), holds in arbitrary domains (not necessarily bounded), and we can assume $(f, g)$ are unique $(P \otimes Q)$-a.s. once we fix $E_P f(X) = E_Q g(Y)$. Notice that if $f_\infty = f, g_\infty = g$, $P$-a.s. and $Q$-a.s. we can conclude: on each compact we apply the above argument starting with any arbitrary subsequence $n_k$ and find a subsequence such that $f_{n_{k_j}} \to f, g_{n_{k_j}} \to g$; therefore $f = \lim f_n(x)$ and $g(y) = \lim g_n$, uniformly in compacts.

It therefore suffices to show that that i) $(f_\infty, g_\infty)$ satisfy the dual optimality conditions and that $f_\infty$ (respectively $g_\infty$) is $P$ (respectively $Q$) integrable, with $E_P f_\infty(X) = E_Q g_\infty(Y)$. Let's prove i. Passing to a subsequence, we assume $f_n \to f$ and $g_n \to g$ uniformly on compact sets. We have

$$e^{-f_\infty(x)} = \lim_{n\to\infty} \int e^{g_n(y) - \frac{1}{2}\|x-y\|^2} \, dQ_n(y)$$

$$e^{-g_\infty(y)} = \lim_{n\to\infty} \int e^{f_n(x) - \frac{1}{2}\|x-y\|^2} \, dP_n(x) \,.$$

It suffices to show that the order of the limit and integral on the right side can be swapped. For a fixed $x$ we observe that Proposition A.1 implies that the integrand is dominated by a uniformly integrable function. Therefore for an arbitrary $\varepsilon > 0$ there exists a compact set $K$ such that

$$\int_{K^C} e^{g_\infty(y) - \frac{1}{2}\|x-y\|^2} \, dQ(y) \leq \varepsilon$$

$$\int_{K^C} e^{g_n(y) - \frac{1}{2}\|x-y\|^2} \, dQ_n(y) \leq \varepsilon \qquad \forall n \geq 0 \,.$$

Write $v_n(y) = e^{g_n(y) - \frac{1}{2}\|x-y\|^2}$ and $v_\infty = e^{g_\infty(y) - \frac{1}{2}\|x-y\|^2}$. Since $g_n$ converges uniformly in compacts so does $v_n$; in particular, there exists $n_0$ such that if $n \geq n_0$,

$$|v_n(y) - v_\infty(y)| \leq \epsilon, \forall y \in \mathcal{K}. \tag{1}$$

Also, since $v_\infty$ is $Q$-integrable, by the strong law of large numbers, almost surely there exists an $n_1$ such that if $n \geq n_1$,

$$\left| \int v_\infty(y) dQ_n(y) - \int v_\infty(y) \, dQ(y) \right| \leq \epsilon, \tag{2}$$

We obtain that for $n$ sufficiently large,

$$\left| \int v_n(y) dQ_n(y) - \int v_\infty(y) \, dQ(y) \right| \leq 4\varepsilon \,.$$

Since $\varepsilon$ was arbitrary, we obtain

$$e^{-f_\infty(x)} = \int v_\infty(y) \, dQ(y) = \int e^{g_\infty(y) - \frac{1}{2}\|x-y\|^2} \, dQ_n(y) \,.$$

Repeating the proof for $g_\infty$, we obtain that $(f_\infty, g_\infty)$ satisfy the dual optimality conditions.

Clearly $(f_\infty, g_\infty)$ are integrable by dominated convergence, and an argument analogous to the one used to show dual optimality establishes that $E_P f_\infty(X) = E_Q g_\infty(Y)$. The claim is therefore proved. $\qquad \square$

**Lemma A.2.** *Suppose $P$ is a $\sigma^2$-subgaussian measure. Then, there exists a (random) $\sigma_u < \infty$ such that $\{P_n\}, P$ are uniformly $\sigma_u^2$-subgaussian $P$ almost surely.*

*Proof.* By definition, there exists $\sigma > 0$ such that $E_P\left(e^{\frac{||X||^2}{2\sigma^2 d}}\right) \leq 2$. By the strong law of large numbers we have that $P$ almost surely

$$\lim_{n\to} E_{P_n}\left(e^{\frac{||X||^2}{2d\sigma^2}}\right) = E_P\left(e^{\frac{||X||^2}{2\sigma^2 d}}\right) \leq 2\,.$$

In particular, this implies the sequence $E_{P_n}\left(e^{\frac{||X||^2}{2\sigma^2 d}}\right)$ is bounded by a random positive number. By the equivalence of definitions of subgaussianity, this implies that $P_n$ are uniformly subgaussian, with a new parameter that we call $\sigma_u^2$. $\qquad\square$

**Proposition A.3.** *Assume $P$ and $Q$ are subgaussian. Let $(f, g)$ be the corresponding optimal dual potentials constructed in Proposition A.1, and define*

$$R_n = S(P_n, Q) - \int f(x)\mathrm{d}P_n(x).$$

*Then,*

$$\lim_{n\to\infty} n\,\mathrm{Var}(R_n) = 0.$$

Our proof relies on the tensorization property for the variance (Efron and Stein, 1981; Boucheron et al., 2013; van Handel, 2014), also known as Efron-Stein inequality: Let $X_1, \ldots X_n$ be i.i.d r.v's with distribution $P$ and $X_1', \ldots X_n'$ be independent copies of $X_1, \ldots X_n$. Also, let $w$ be an arbitrary measurable function of the sample that is symmetric on its coordinates, and define $Z = \omega\left(X_1, \ldots X_n\right)$ and $Z' = \omega\left(X_1', X_2, \ldots X_n\right)$. Then,

$$Var(Z) \leq \frac{n}{2} E(Z - Z')_+^2. \tag{3}$$

*Proof of Proposition A.3.* Denote by $P_n'$ the empirical distribution of $X_1', X_2, \ldots X_n$, and let

$$R_n' = S(P_n', Q) - \int f(x)\mathrm{d}P_n'(x).$$

by Efron-Stein, it suffices to show $\lim_{n\to\infty} n^2 E(R_n - R_n')_+^2 = 0$. We divide the proof in the verification of two statements. First, we show $\lim_{n\to\infty} n(R_n - R_n')_+ = 0$. We will then show that $n^2(R_n - R_n')_+^2$ is uniformly integrable.

Call $(f_n, g_n)$ the optimal potentials associated to $(P_n, Q)$. Since $P_n$ is subgaussian by Lemma A.2, Proposition A.1 implies that we can assume that $(f_n, g_n)$ satisfy the dual optimality conditions for all $x, y \in \mathbb{R}^d$. Therefore

$$S(P_n, Q) = \int f_n(x)\,\mathrm{d}P_n(x) + \int g_n(y)\,\mathrm{d}Q(y),$$

$$S(P_n', Q) \geq \int f_n(x)\mathrm{d}P_n'(x) + \int g_n(y)\mathrm{d}Q(y) - \iint e^{f_n(x)+g_n(y)-\frac{1}{2}||x-y||^2}\mathrm{d}P_n'(x)\mathrm{d}Q(y) + 1$$

$$= \int f_n(x)\,\mathrm{d}P_n'(x) + \int g_n(y)\,\mathrm{d}Q(y)\,.$$

Therefore,

$$n(R_n - R_n')_+ \leq (f_n(X_1) - f(X_1)) - (f_n(X_1') - f(X_1'))\,.$$

By Proposition A.2, $(f_n, g_n)$ converges pointwise to $(f, g)$ almost surely, so $\lim_{n\to\infty} n(R_n - R_n')_+ = 0$ almost surely.

To show uniform integrability, we note that $n(R_n - R_n') = n(S(P_n, Q) - S(P_n', Q)) - (f(X_1) - f(X_1'))$ and by Proposition A.1 and the subgaussianity of $P$, $f(X_1), f(X_1')$ have finite second moments. It therefore suffices to show that $n^2(S(P_n, Q) - S(P_n', Q))_+^2$ is uniformly integrable.

Let $\pi'$ be the underlying optimal entropic coupling between $P'_n$ and $Q$ that we disintegrate in terms of $Q$ and the (random) kernel $\{P'(\cdot|y)\}_y$ of conditional distributions over the sample $P'_n$ given $y$, i.e.

$$\mathrm{d}\pi'(x,y) = \mathrm{d}Q(y)\left(P'(x|y)\delta_{X'_1}(x) + \sum_{i=2}^{n} P'(x|y)\delta_{X_i}(x)\right).$$

We now slightly modify $\pi'$ to make it have $P_n$ as first marginal; specifically, we define

$$\mathrm{d}\bar{\pi}(x,y) = \mathrm{d}Q(y)\left(\sum_{i=1}^{n} \bar{P}(x|y)\delta_{X_i}(x)\right), \text{ with } \bar{P}(x|y) = \begin{cases} P'(X'_1|y) & x = X_1 \\ P'(X_i|y) & x = X_i, i \neq 1 \end{cases}.$$

By the definitions of $S(P_n, Q)$ and $S(P'_n, Q)$, it is easily verified that

$$S(P_n, Q) \leq \sum_{i=1}^{n} \int \frac{\|X_i - y\|^2}{2}\bar{P}(X_i|y)\mathrm{d}Q(y) + I(\bar{\pi}),$$

and that

$$S(P'_n, Q) = \int \frac{\|X'_1 - y\|^2}{2}P'(X'_1|y)\mathrm{d}Q(y) + \sum_{i=2}^{n} \int \frac{\|X_i - y\|^2}{2}P'(X_i|y)\mathrm{d}Q(y) + I(\pi'),$$

where $I(\cdot)$ denotes mutual information. Therefore,

$$S(P_n, Q) - S(P'_n, Q) \leq I(\bar{\pi}) - I(\pi') + \int \frac{\|X_1 - y\|^2 - \|X'_1 - y\|^2}{2}P'(X'_1|y)\mathrm{d}Q(y). \quad (4)$$

Observe that $I(\bar{\pi}) = I(\pi')$ since $I(\pi')$ doesn't depend on the sample values, but only in the way the conditionals $P'(\cdot|y)$ split over the sample, which by construction is the same for both $\bar{\pi}$ and $\pi'$. Therefore, we only need to bound the (expected squared) integral in (4), and we proceed as in Del Barrio and Loubes (2019). Specifically, we have

$$\begin{aligned} S(P_n, Q) - S(P'_n, Q) &\leq \int \frac{\|X_1 - y\|^2 - \|X'_1 - y\|^2}{2}P'(X'_1|y)\mathrm{d}Q(y) &(5) \\ &\leq \frac{1}{2}\|X_1 - X'_1\|\left(\frac{\|X_1\| + \|X'_1\|}{n} + 2\int \|y\|\, P'(X'_1|y)\mathrm{d}Q(y)\right), \end{aligned}$$

from which it follows that

$$n^2(S(P_n, Q) - S(P'_n, Q))_+^2 \leq (\|X_1 - X'_1\|^2 \|X_1\|^2) + n^2\|X_1 - X'_1\|^2\left(\int \|y\|\, P'(X'_1|y)\mathrm{d}Q(y)\right)^2. \quad (6)$$

The first term is clearly uniformly integrable since $P$ has moments of all orders, so we focus on the second term.

By Cauchy-Schwartz,

$$\begin{aligned} E\left(\|X_1 - X'_1\|^4\left(\int \|y\|\, P'(X'_1|y)\mathrm{d}Q(y)\right)^4\right)^2 &\leq E\left(\|X_1 - X'_1\|^8\right) \times \\ &\qquad E\left(\left(\int \|y\|\, P'(X'_1|y)\mathrm{d}Q(y)\right)^8\right). \end{aligned}$$

And now, by Hölder's inequality,

$$\begin{aligned} \left(\int \|y\|\, P'(X'_1|y)\mathrm{d}Q(y)\right)^8 &\leq \left(\int P'(X'_1|y)\mathrm{d}Q(y)\right)^7\left(\int \|y\|^8\, P'(X'_1|y)\mathrm{d}Q(y)\right) \\ &= \frac{1}{n^7}\left(\int \|y\|^8\, P'(X'_1|y)\mathrm{d}Q(y)\right). \end{aligned}$$

Also, notice that the r.v's $\int \|y\|^8 P'(X_i'|y)\mathrm{d}Q(y)$ are equally distributed, and therefore

$$E\left(\int \|y\|^8 P'(X_1'|y)\mathrm{d}Q(y)\right) = \frac{1}{n}E\left(\sum_{i=1}^n \int \|y\|^8 P'(X_i'|y)\mathrm{d}Q(y)\right) = \frac{1}{n}E\left(\int \|y\|^8 \mathrm{d}Q(y)\right).$$

We obtain

$$E\left(\left(\int \|y\| P'(X_1'|y)\mathrm{d}Q(y)\right)\right)^8 \leq \frac{1}{n^8}\int \|y\|^8 \mathrm{d}Q(y). \tag{7}$$

Together, (7) and(7) imply that the quantity $n^2\|X_1 - X_1'\|^2 \left(\int \|y\| P'(X_1'|y)\mathrm{d}Q(y)\right)^2$ has uniformly bounded second moments, and is therefore uniformly integrable. Therefore $n^2(S(P_n,Q) - S(P_n',Q))_+^2$ is uniformly integrable as well, and combining this with the almost sure convergence implies the claim. $\qquad\square$

**Lemma A.3.** *Let $\mu_\beta$ be defined as in* (8). *Then*

$$|\mu_\beta| \leq C_{|\beta|,d}\begin{cases} \sigma^{|\beta|}(\sigma + \sigma^2)^{|\beta|} & \|x\| \leq \sqrt{d}\sigma \\ \sigma^{|\beta|}(\sqrt{\sigma\|x\|} + \sigma\|x\|)^{|\beta|} & \|x\| > \sqrt{d}\sigma. \end{cases}$$

*Proof.* To bound $\mu_\beta$, we split the integral in the numerator according to the norm of $y$. Let $A = \{y : \|y\| \leq \tau\}$, where $\tau$ is a threshold to be chosen. Then

$$\mu_\beta = \frac{\int \mathbb{1}_A y^\beta e^{g(y)-\frac{1}{2}\|y\|^2+x\cdot y}\,\mathrm{d}Q(y)}{\int e^{g(y)-\frac{1}{2}\|y\|^2+x\cdot y}\,\mathrm{d}Q(y)} + \frac{\int \mathbb{1}_{\overline{A}} y^\beta e^{g(y)-\frac{1}{2}\|y\|^2+x\cdot y}\,\mathrm{d}Q(y)}{\int e^{g(y)-\frac{1}{2}\|y\|^2+x\cdot y}\,\mathrm{d}Q(y)}.$$

The first term is clearly bounded by $\tau^\beta$. For the second, we apply Proposition A.1 to show

$$\left(\int e^{g(y)-\frac{1}{2}\|y\|^2+x\cdot y}\,\mathrm{d}Q(y)\right)^{-1} = e^{-\frac{1}{2}\|x\|^2}e^{f(x)} \leq e^{d\sigma^2+\sqrt{d}\sigma\|x\|}$$

and

$$e^{g(y)-\frac{1}{2}\|y\|^2} \leq e^{d\sigma^2+\sqrt{d}\sigma\|y\|}.$$

We obtain

$$\frac{\int \mathbb{1}_{\overline{A}} y^\beta e^{g(y)-\frac{1}{2}\|y\|^2+x\cdot y}\,\mathrm{d}Q(y)}{\int e^{g(y)-\frac{1}{2}\|y\|^2+x\cdot y}\,\mathrm{d}Q(y)} \leq e^{2d\sigma^2+\sqrt{d}\sigma\|x\|}\int \mathbb{1}_{\overline{A}} y^\beta e^{\sqrt{d}\sigma\|y\|+x\cdot y}\,\mathrm{d}Q(y)$$

$$\leq e^{2d\sigma^2+\sqrt{d}\sigma\|x\|}\left(\int \mathbb{1}_{\overline{A}} y^{2\beta}\,\mathrm{d}Q(y)\right)^{1/2}\left(\int e^{2(\sqrt{d}\sigma+\|x\|)\|y\|}\,\mathrm{d}Q(y)\right)^{1/2}$$

Since $Q$ is subgaussian, Lemma A.1 and the definition of $A$ imply

$$\left(\int \mathbb{1}_{\overline{A}} y^{2\beta}\,\mathrm{d}Q(y)\right)^{1/2} \leq e^{-\frac{\tau^2}{8d\sigma^2}}\left(\int e^{\frac{\|y\|^2}{4d\sigma^2}}y^{2\beta}\,\mathrm{d}Q(y)\right)^{1/2} \leq \sqrt{2}e^{-\frac{\tau^2}{8d\sigma^2}}(2|\beta|)!^{1/4}(\sqrt{2d}\sigma)^{|\beta|}.$$

Lemma A.1 also implies

$$\int e^{2(\sqrt{d}\sigma+\|x\|)\|y\|}\,\mathrm{d}Q(y) \leq 2e^{2d\sigma^2(\|x\|+\sqrt{d}\sigma)^2}.$$

Therefore, if we choose $\tau^2 \geq C_{|\beta|,d}(\sigma^4 + \sigma^6)$ if $\|x\| \leq \sqrt{d}\sigma$ and $\tau^2 \geq C_{|\beta|,d}(\sigma^3\|x\| + \sigma^4\|x\|^2)$ if $\|x\| > \sqrt{d}\sigma$ for a sufficiently large constant $C_{|\beta|,d}$, then we will have

$$\frac{\int \mathbb{1}_{\overline{A}} y^\beta e^{g(y)-\frac{1}{2}\|y\|^2+x\cdot y}\,\mathrm{d}Q(y)}{\int e^{g(y)-\frac{1}{2}\|y\|^2+x\cdot y}\,\mathrm{d}Q(y)} \leq C_{|\beta|,d}(\sqrt{d}\sigma)^{|\beta|}$$

Combining this with the bound on the first term yields the claim. $\qquad\square$

**Lemma A.4.** *Let $\tilde\sigma$ be defined as in the proof of Theorem 2. Then for any positive integer $k$,*

$$E\tilde\sigma^{2k} \leq 2k^k\sigma^{2k}.$$

*Proof.* First, let $P$ be an arbitrary probability distribution, and let $\alpha > 0$. We first show that if $t = E_P e^{\frac{\|X\|^2}{\alpha}}$ is finite, then $P$ is $t\frac{\alpha}{2d}$-subgaussian. To see this, set $\tau^2 = t\frac{\alpha}{2d}$. Then

$$Ee^{\frac{\|X\|^2}{2d\tau^2}} \leq \left(Ee^{\frac{\|X\|^2}{\alpha}}\right)^{\frac{\alpha}{2d\tau^2}} = t^{1/t} \leq e^{1/e} < 2\,,$$

where the first step uses Jensen's inequality and the fact that $t \geq 1$.

The above considerations imply that if $Q$ is $\sigma^2$ subgaussian and we set

$$\tau^2 = \max\{E_{P_n} e^{\frac{\|X\|^2}{2kd\sigma^2}} k\sigma^2, E_{Q_n} e^{\frac{\|Y\|^2}{2kd\sigma^2}} k\sigma^2\}\,,$$

then $P_n$, $Q_n$, $P$, and $Q$ are all $\tau^2$ subgaussian, which implies that $\tilde{\sigma}^2 \leq \tau^2$. Therefore, by Jensen's inequality,

$$\tilde{\sigma}^{2k} \leq E_{P_n} e^{\frac{\|X\|^2}{2d\sigma^2}} k^k \sigma^{2k} + E_{Q_n} e^{\frac{\|Y\|^2}{2d\sigma^2}} k^k \sigma^{2k}\,,$$

and taking expectations with respect to $P$ and $Q$ yields

$$E\tilde{\sigma}^{2k} \leq E_P e^{\frac{\|X\|^2}{2d\sigma^2}} k^k \sigma^{2k} + E_Q e^{\frac{\|Y\|^2}{2d\sigma^2}} k^k \sigma^{2k} \leq 4k^k \sigma^{2k}\,.$$

$\square$

# B  Omitted proofs

## B.1  Proof of Proposition 1

We choose the potentials $f$ and $g$ as in Proposition A.1. That establishes the $k = 0$ case.

For convenience, write $\overline{f}(x) = f(x) - \frac{1}{2}\|x\|^2$. We seek to bound $|D^\alpha \overline{f}(x)|$.

Our calculation is similar to classical calculations which relate the cumulants of a distribution to its moments (see McCullagh, 1987, Section 2.3). Given a multi-index $\beta$, write

$$\mu_\beta = \frac{\int y^\beta e^{g(y) - \frac{1}{2}\|y\|^2 + x \cdot y} \, \mathrm{d}Q(y)}{\int e^{g(y) - \frac{1}{2}\|y\|^2 + x \cdot y} \, \mathrm{d}Q(y)}\,. \tag{8}$$

We use the convention that $y^\beta = \prod_{i=1}^d y_i^{\beta_i}$. The notation $\mu_\beta$ is chosen to remind the reader that these quantities are moments of $y$ under the tilted measure whose density with respect to $Q$ is proportional to $e^{g(y) - \frac{1}{2}\|y\|^2 + x \cdot y}$.

By the multivariate Faá di Bruno formula (see, e.g. Constantine and Savits, 1996),

$$D^\alpha \overline{f}(x) = -D^\alpha \log(e^{-\overline{f}(x)}) = \sum_{\substack{\beta_1, \dots, \beta_k \\ \beta_1 + \dots + \beta_k = \alpha}} \lambda_{\alpha, \beta_1, \dots, \beta_k} \prod_{j=1}^k \mu_{\beta_j}\,, \tag{9}$$

where the coefficients $\lambda_{\alpha, \beta_1, \dots, \beta_k}$ are combinatorial quantities related to partitions of $[k]$ whose precise value is unimportant.

Applying Lemma A.3 yields the claim.

## B.2  Proof of Proposition 3

We use the symbol $C$, decorated with subscripts, to indicate constants whose value may change from line to line. We apply van der Vaart and Wellner (1996, Corollary 2.7.4). Denote by $L$ the quantity $\frac{1}{n}\sum_{i=1}^n e^{\|x_i\|^2/2d\sigma^2}$. The subgaussianity of $P$ implies that $EL \leq 2$. We partition $\mathbb{R}^d$ into sets $B_j$ defined by $B_0 = [-\sigma, \sigma]^d$ and $B_j = [-2^j\sigma, 2^j\sigma] \setminus [-2^{j-1}\sigma, 2^{j-1}\sigma]$. Note that for each $j$, the Lebesgue measure of $\{x : \mathrm{d}(x, B_j) \leq 1\}$ is bounded by $C_d(1 + \sigma^d 2^{dj})$. Moreover, by Markov's inequality, the mass that $P_n$ assigns to each $B_j$ is at most $Le^{-2^{2j-3}}$. Finally, by definition of the class $\mathcal{F}^s$, the functions in $\mathcal{F}^s$ have $\mathcal{C}^s(B_0)$ norm at most $C_{s,d}(1 + \sigma^s)$, and on $B_j$ for $j \geq 1$ have $\mathcal{C}^s(B_j)$ norm at most $C_{s,d}2^{js}(1 + \sigma^s)$, where $\mathcal{C}^s(\Omega)$ represents the Hölder space on $\Omega$ of smoothness $s$.

Applying van der Vaart and Wellner (1996, Corollary 2.7.4) with $V = d/s$ and $r = 2$ yields

$$\log N(\varepsilon, \mathcal{F}^s, L_2(P_n)) \leq C_d \varepsilon^{-d/s} L^{d/2s} \left( \sum_{j \geq 0} (1 + \sigma^d 2^{dj})^{\frac{2s}{d+2s}} 2^{\frac{2djs}{d+2s}} (1 + \sigma^s)^{\frac{2d}{d+2s}} e^{-\frac{d2^{2j-3}}{d+2s}} \right)^{\frac{d+2s}{2s}}$$

$$\leq C_d \varepsilon^{-d/s} L^{d/2s} (1 + \sigma^{2d}) \left( \sum_{j \geq 0} 2^{\frac{4djs}{d+2s}} e^{-\frac{d2^{2j-3}}{d+2s}} \right)^{\frac{d+2s}{2s}}$$

$$\leq C_d \varepsilon^{-d/s} L^{d/2s} (1 + \sigma^{2d}),$$

where the final step follows because the series is summable with value independent of $\sigma$ and $L$.

To show the second claim, we note that $E_{P_n} \|X\|^4 \leq C_d L \sigma^4$ by the same argument used to bound the moments of $P$ in Lemma A.1. The definition of the class $\mathcal{F}^s$ implies

$$\max_{f \in \mathcal{F}^s} \|f\|_{L_2(P_n)}^2 = \max_{f \in \mathcal{F}^s} E_{P_n} |f(X)|^2 \leq C_d E_{P_n} (1 + \|X\|^4) \leq C_d (1 + L\sigma^4).$$

$\square$