[Reviews · NeurIPS 2019]

Reviewer 1



This paper provides several improved results on statistical qualities of entropic optimal transport. Entropic OT has been a heated area of research for the past several years. This paper provides improved theoretical understanding for entropic OT from a statistical point of view, improving upon previous bounds and providing a central limit theorem for entropic OT for subgaussian random variables. Previous CLT was known only for for probability measures supported on a finite set of points. Originality: Provides fresh theoretical results in sample complexity and limiting distribution that improve upon previously known results. Quality: The problems are well-motivated and placed in context. Theoretical claims are supported by proofs. Simulation studies assess empirical performance. Clarity: The paper is well-written and clear. Significance: This paper provides improved statistical understanding of entropic OT, with a highly improved sample complexity bound, and more broadly applicable CLT. *** I thank the authors for their careful reading of our reviews and for addressing my concerns.

Reviewer 2



AFTER REBUTTAL I thank the authors for their careful consideration of our remarks, I believe the paper will benefit from all these comments and am confident the authors will be able to improve it accordingly. ____________________________________________ The theoretical contributions of this paper for entropy-regularized OT are : - getting rid of a bad exponential factor in the previous sample complexity bound by Genevay et al - extending that result to unbounded domains for sub-gaussian distributions - a CLT for entropy-regularized OT, based on the proof for the standard case. The theoretical part is overall clear and well written, although: - briefly introducing F_sigma would improve overall readability (e.g introducing it as the set of functions with bounded successive derivatives) - it would be nice to state more clearly (in the abstract or intro) which additional hypotheses you use to improve existing results (subgaussianity + quadratic cost hypothesis). I believe it is not essential to get rid of the exponential bound, but required for the extension to unbounded domains? A couple of typos: - line 136 : should be v instead of u in the second term - line 140 : should be v instead of u - line 166 : theorem 3 instead of 2 Regarding the application to entropy estimation, I feel that it would benefit from a little rewriting and some clarifications in the experimental section. - prop 3 : what is Z_g? I an guessing the normalization constant for phi_g. - equation 13 : I understand here that S is the entropic OT where the cost function is c(x,y) = g(x-y) but it should be stated more clearly (maybe introduce S_c to insist on the dependence of S on the cost function). In the experiments, in figure 1, should the legend read epsilon instead of sigma_g? I am confused.

Reviewer 3



Typo: Just below (2) defiendiswriensteadofdefined'. There are many places in which the notation c(x,y) is used without it being defined, for example in (3). By comparing with Proposition 3 much later in the paper it seems that the authors intend c(x,y)=\|x-y\|^2/2, but this is not at all clear earlier in the paper. At the top of page 3 the authors state that the Wasserstein distance W_2^2(P_n,Q_n) converges to W_2^2(P,Q) no faster than n^{-1/d}. A citation should be provided. There is no discussion in the paper of how close S_\epsilon is to W_2^2, or which values of \epsilon one is interested in in practice. Is the idea to choose \epsilon small in order to estimate W_2? In this case I would have liked to see some bounds on |S_\epsilon - W_2^2|. Page 5, line 139. The authors write `Applying the bound on E L^2...', but no such bound has been established. Do they mean the bound on E L given in Proposition 2? This would suffice. It is not made clear what the functions f and g are in the statement of Theorem 3. I guess they are the optimal potentials from (4), but this should be stated. On page 6 the authors write that the proof of Theorem 2 has two main steps, and they label these steps (a) and (c). Is there a step (b) that is missing? Or did they mean to write (a) and (b)? My main concern is with the proof of Proposition 3 that provides the link between EOT and entropy estimation. It is written that \nu is assumed to be a translation invariant measure. In the proof, \beta is a probability measure which, in line 195, is taken to be \nu, and so it must be the case that \nu can be taken to be a probability measure. However, it seems to me that the only translation invariant measure on \mathbb{R}^d is given by the Lebesgue measure (up to a scale factor), which cannot be normalized to be a probability measure on all of \mathbb{R}^d. Therefore, it seems necessary to take \nu to be compactly supported, which is not what the authors are focussing their attention on, as they claim that there results only require bounds on the subgaussian constant, and not on the support. The proof of Propostion 3 thus seems to be incomplete. ========================================================== UPDATE (after author rebuttal): The authors' rebuttal addressed most of my concerns. In reply to the rebuttal, I will just add that Theorem 4 on entropy estimation here does not show how the bounds depend on the dimension d. If you want to discuss the behaviour of the entropy estimator when d is large then the bounds should show this dependence. I have now changed the overall score.

[Author Response · NeurIPS 2019]

1 We thank the reviewers for their professional work, constructive and thorough criticism, and sensible directions for
2 improvement. Altogether, their judgement suggests an acceptance. Below we elaborate a response on the main criticisms
3 that we will incorporate on strengthened final version.
4 **Typos and lack of consistency/clarity:**

- **Reviewer 2** pointed to typos in lines 136, 140, 166, lack of clarity in the definition of $Z_g$ and lack of clarity in the definition of the entropic cost in terms of a generic cost function $c(x, y)$. We fully agree with the criticism and will improve the manuscript based on it.

- **Reviewer 2** points to an inconsistency in Figure 1. There, both $\epsilon$ and $\sigma_g^2$ can be used in the legend but we will stick to one to keep notation simple. She also suggests introducing $\mathcal{F}_\sigma$ earlier to improve readability, and we will do so.

- **Reviewer 2 Reviewer 4** points there is a reference to a bound for $EL^2$ that is not stated elsewhere. There, we meant $EL$ and the bound is the one of Proposition 2, as **Reviewer 4** guessed.

- **Reviewer 4** points to a missing (b) in page 6. Indeed, (c) should have been (b). We will correct this.

- **Reviewer 4** points it is not clear that $(f, g)$ in Theorem 3 are the optimal potentials. We will make clear whenever we write $(f, g)$ we are referring to them.

16 **Hypothesis: Reviewer 2** suggest being finer in stating which hypothesis are required for each of our improvements.
17 Indeed, subgaussianity is needed for the extension to unbounded domains, but the exponential factor improvement is a
18 separate contribution that would hold under the same hyphotesis of Genevay et al. (2019). We will make this clear.

19 **Error in proof of theorem 3**: We thank **Reviewer 4** for pointing out. We do agree there is an error in the proof, but it
20 can be easily fixed. It can be shown that as long as $\beta$ is $\sigma$-finite it is still possible to make sense of the relative entropy
21 as if $\beta$ was a probability measure. This problem has been addressed by e.g. Léonard (2014): essentially, we could
22 simply define the relative entropy $H(\alpha|\beta)$ for unbounded $\beta$ by the usual formula (line 55), and then what remains to
23 verify is that we can still go from the definition in equation (14) to equations (15) and (16). In Léonard (2014) it is
24 argued all measure theoretical subtleties can be sorted out. We will fix the prove by comment on this.

25 **Lack of distributional limits**: **Reviewer 4** pointed out a weakness of our CLT is that it expresses in terms of the
26 centering constants $E(S(P_n, Q_n))$ instead of $S(P, Q)$. We agree it is a weakness, and we don't know if this can be
27 improved, which boils down to answering whether the sample complexity $O(1/\sqrt{n})$ could be indeed $o(1/\sqrt{n})$. We will
28 comment on this issue. Nonetheless, in practice it is observed that regardless of the rate in $n$, dependence on dimension
29 can be bad (indeed, exponential, as shown in Goldfeld et al. (2019) for entropy estimation), making the construction of
30 confidence intervals hard, even if the CLT holds with $S(P, Q)$ instead of $E(S(P_n, Q_n))$.

31 **Other entropy estimators**: **Reviewer 4** points to the recent work of Berrett et al. (2019) showing $\sqrt{n}$ consistent
32 estimators for a certain class of smooth densities, with available distributional limits. We will cite this relevant work,
33 but it isn't clear Berrett et al. (2019) is superseding our method, as for the CLT of Berrett et al. (2019) to kick in, it is
34 required that a certain parameter that the number of neighbors $k$ is greater than a constant that may depends wildly on
35 dimension. It is shown (empirically) in Goldfeld et al. (2019) that their estimator (and hence, ours) outperforms the one
36 of Berrett et al. (2019) in practice. Finally, it is unclear whether the method of Berrett et al. (2019) can be applied to all
37 distributions in our setup; only for compactly supported $P$ the inclusion is straightforward.

38 **Comments on lower bounds, optimal constant as a function of subgaussian constant, heavier tailed distributions**:
39 **Reviewers 1 and 4** mentioned several improvement directions related to sharper statements of bounds, and extending
40 them to heavier tailed distributions. We recognize them as valuable directions, and at least we will comment on these
41 issues. However, we don't compromise for a definite answer of this issues but rather defer them as future work. In
42 particular, we will try to provide lower bounds, e.g. arguing as in Goldfeld et al. (2019) and explore other cases beyond
43 subgaussianity. Empirical results suggest they are valid in heavier tailed cases as well, but the extension is nontrivial
44 because our machinery relies heavily in subgaussianity.

45 # References

46 Berrett, T. B., Samworth, R. J., Yuan, M., et al. (2019). Efficient multivariate entropy estimation via $k$-nearest neighbour
47 distances. *The Annals of Statistics*, 47(1):288–318.

48 Genevay, A., Chizat, L., Bach, F., , Cuturi, M., and Peyré, G. (2019). Sample complexity of sinkhorn divergences. In
49 *Proceedings of the 22nd International Conference on Artificial Intelligence and Statistics (AISTATS)*.

50 Goldfeld, Z., Greenewald, K., Polyanskiy, Y., and Weed, J. (2019). Convergence of smoothed empirical measures with
51 applications to entropy estimation. *arXiv preprint arXiv:1905.13576*.

52 Léonard, C. (2014). Some properties of path measures. In *Séminaire de Probabilités XLVI*, pages 207–230. Springer.


[Meta-Review · NeurIPS 2019]

The reviewers found the contributions of the paper novel and significant. The author rebuttal helped clarify a few issues raised initially.